Methods

# Patient-specific analysis of co-expression to measure biological network rewiring in individuals

Lanying Wei , Yucui Xin, Mengchen Pu , Yingsheng Zhang

To effectively understand the underlying mechanisms of disease and inform the development of personalized therapies, it is critical to harness the power of differential co-expression (DCE) network analysis. Despite the promise of DCE network analysis in precision medicine, current approaches have a major limitation: they measure an average differential network across multiple samples, which means the specific etiology of individual patients is often overlooked. To address this, we present Cosinet, a DCE-based single-sample network rewiring degree quantification tool. By analyzing two breast cancer datasets, we demonstrate that Cosinet can identify important differences in gene co-expression patterns between individual patients and generate scores for each individual that are significantly associated with overall survival, recurrence-free interval, and other clinical outcomes, even after adjusting for risk factors such as age, tumor size, HER2 status, and PAM50 subtypes. Cosinet represents a remarkable development toward unlocking the potential of DCE analysis in the context of precision medicine.

## Introduction

Differential co-expression (DCE) network analysis is a valuable approach for understanding the gene regulation of disease and guiding the development of personalized therapies (Mousavian et al, 2017; Voigt et al, 2017; Luo et al, 2018; van Dam et al, 2018; Savino et al, 2020; Upadhyaya et al, 2020; Yu et al, 2021). It involves constructing networks of co-expressed genes in given sets of samples and comparing these networks between different conditions or treatments (van Dam et al, 2018; Savino et al, 2020). In contrast to differential expression analysis, which aims to identify genes that are expressed at different levels between two or more conditions or treatments (McDermaid et al, 2019), this approach allows examination of relationships between genes, providing a more comprehensive view of changes in gene expression patterns (Xu et al, 2016; van Dam et al, 2018; Bhuva et al, 2019; Savino et al, 2020). Furthermore, correlation changes between genes can occur without affecting their expression levels, suggesting some regulatory changes may be overlooked by traditional differential gene expression analysis (Xu et al, 2016; Bhuva et al, 2019). Through the comparison of co-expression networks between healthy and disease samples, researchers can identify genes that are differentially coordinated between the two conditions, as well as pathways or processes that are disrupted or rewired in the disease state (Xu et al, 2016; Mousavian et al, 2017; Voigt et al, 2017; Luo et al, 2018; van Dam et al, 2018; Bhuva et al, 2019; Savino et al, 2020; Upadhyaya et al, 2020; Yu et al, 2021). By understanding the changes in gene co-expression patterns that are associated with a particular disease, researchers can gain insights into the underlying mechanisms of the disease and develop more targeted and effective therapies (Xu et al, 2016; Mousavian et al, 2017; Luo et al, 2018; van Dam et al, 2018; Upadhyaya et al, 2020; Hasankhani et al, 2021).

Despite its potential value, DCE analysis has faced a significant challenge in its application in the field of precision medicine. To analyze a DCE network, at least dozens of samples are required in order to accurately quantify the differences between them (Watson, 2006; Freudenberg et al, 2010; Tesson et al, 2010; Ma et al, 2011; Dawson & Kendziorski, 2012; Amar et al, 2013; Fukushima, 2013; Hsiao et al, 2016; McKenzie et al, 2016; Siska et al, 2016; Tian et al, 2016), and what is measured is the average change in the co-expression network. Recent advancements in this field, exemplified by sample-specific gene interaction perturbations measured by Chen et al (2021) and co-expression network reconstruction methods on single samples as discussed by Guo et al (2020), have underscored the need to examine individual-specific gene interactions for personalized therapy applications. However, to our best knowledge, there is currently no tool that can further quantify the degree of the differences for a single sample in the context of the DCE network. This brings a limitation in the context of precision medicine, where the goal is to tailor a treatment to the specific characteristics of an individual patient (Ashley, 2016; Dugger et al, 2018). In order to fully realize the potential of precision medicine, it is critical to be able to further analyze and interpret gene co-expression patterns at the individual level.

In response to this limitation, we present Cosinet, the first DCE-based single-sample network rewiring degree quantification tool. Cosinet uses gene expression data to determine the degree of

Beijing StoneWise Technology Co Ltd, Danling SOHO, Beijing, China

Correspondence: weilanying@stonewise.cn; zhangyingsheng@stonewise.cn

similarity between the gene co-expression patterns of an individual sample and reference conditions in the context of a function-specific DCE network. It employs a combination of techniques such as DCE analysis, network centrality calculation, gene set enrichment analysis (GSEA), and a novel statistic developed by our team to measure the differences in the statistical independence of a gene pair between two conditions. To demonstrate the potential of Cosinet in personalized therapy decisions, we performed an analysis using data from breast cancer patients. Quantifying the degree of rewiring in terms of the estrogen receptor (ER) response network for each ER-positive (ER+) sample relative to the whole ER+ and ER-negative (ER−) groups, we found that the quantified scores were significantly associated with survival outcomes in ER+ samples treated with adjuvant endocrine therapy. Specifically, after adjustment for age, tumor size, and HER2 status, a one-point increase in the Cosinet score reduced the hazard of death by 49% (95% CI: 33.7–60.5%). In addition, the association was verified in a separate validation dataset, which confirmed the significant association between the Cosinet score and key clinical endpoints such as overall survival, recurrence-free interval (RFi), and other critical endpoints, even after adjusting for multiple relevant covariates. These findings underscore the potential of the Cosinet score as a valuable biomarker to inform personalized treatment decisions.

# Results

## The workflow of Cosinet

The general Cosinet workflow (Fig 1) consists of the following steps: (1) construct a global DCE network between two treatments or conditions using a z-score–based method (Bhuva et al, 2019). Applying the Fisher transformation to the correlation coefficients and conducting a subsequent z test, we compute a z-score matrix representing the global DCE network. (2) Calculate eigenvector centralities for nodes in the network, and rank genes based on centrality. This allows us to identify genes involved in highly rewired sub-networks. (3) Perform GSEA to identify function- or pathway-specific gene sets whose members are overrepresented at the top of the ranked list. The gene co-expression patterns of significantly enriched pathways or functions that are highly rewired between the two compared conditions indicate that the corresponding biological process may be functionally associated with the specific treatment or condition. (4) Construct function-specific DCE sub-networks. For a gene set that is of research interest, we use the core enrichment genes as nodes, and its gene pairs that show strong changes in gene–gene association as edges, to construct a function-specific DCE sub-network. (5) Compute Cosinet scores using DCE sub-networks. The calculation is based on a statistic $\rho$ originally developed by Dai et al (2019) to construct cell-specific networks (CSNs) from single-cell RNA-sequencing data. They derived $\rho$ as a local estimate of the statistical independence of gene pairs on the basis of probability theory. We use this statistic to measure the statistical independence of a gene pair in a given sample using bulk RNA-sequencing data and modify it to fit our research scenario. The calculation involves determining $P$-values

for both conditions and subtracting them to obtain Δ, an estimator of the rewiring degree of the co-expression pattern for a single sample. The final Cosinet score is calculated by aggregating the Δ values at the sub-network level. A negative Cosinet score signifies that the gene co-expression patterns of the sub-network in a given sample align with those in the first condition. Conversely, a positive Cosinet score indicates that they are more similar to those in the second condition. The magnitude of the Cosinet score indicates the degree of similarity or dissimilarity to a particular condition. More details about the workflow can be found in the Materials and Methods section.

## The effectiveness of the Cosinet algorithm

To test the effectiveness of our algorithm in distinguishing different co-expression patterns, we applied the Cosinet algorithm to the whole 241 ER− and the 241 randomly selected ER+ samples from a publicly available RNA-seq dataset of breast cancer patients (Brueffer et al, 2018) (GEO accession: GSE96058). We compared ER+ (condition 2) with ER− (condition 1) samples to obtain a global DCE network. Gene pairs with substantial alteration in correlations and representative co-expression patterns were extracted, and Cosinet scores were calculated for each individual gene pair without the aggregation step. The results are displayed in scatter plots (Fig 2). Based on the quantified scores, samples whose co-expression pattern aligns with their annotated condition are indicated by a circle, whereas those with inconsistent patterns are indicated by a cross. We can observe that some samples exhibit co-expression patterns predominant in the contrasting condition, implying that their co-expression relationship in certain gene pairs aligns more closely with the opposing case. This observation may be due to misclassification of the samples or more likely from other factors that affect the gene expression profiles, resulting in a pattern that differs from their designated condition. Some of the potential factors are the heterogeneous nature of the disease, the activation or repression of signaling pathways, genetic variation, and epigenetic modifications (Cancer Genome Atlas Network, 2012; Curtis et al, 2012). All of these factors can lead to discordance between the ER status of a sample and its observed co-expression pattern, highlighting the intricate molecular mechanisms underlying the biological process in particular clinical subtypes. Fig 2 demonstrates the effectiveness of the Cosinet algorithm in detecting these discordant samples, both in cases where the co-expression has simple linear relationships, such as linear positive or negative correlations, and in cases with complex co-expression patterns, such as L-shape, reversed-L-shape, N-shape, and X-shape. Notably, downsampling of the ER+ group was designed to enhance visualization clarity, as the excessive number of samples in the ER+ group (n = 2,832) led to point overlap, posing difficulties for clear visualization. The results using the downsampled dataset closely aligned with those obtained from the entire dataset, as depicted in Fig S1.

## Comparison of Cosinet with existing tools for DCE analysis

The calculation of the Cosinet score is based on a given network structure that reflects DCE. Three main approaches are used to construct such networks, each focusing on a different level of

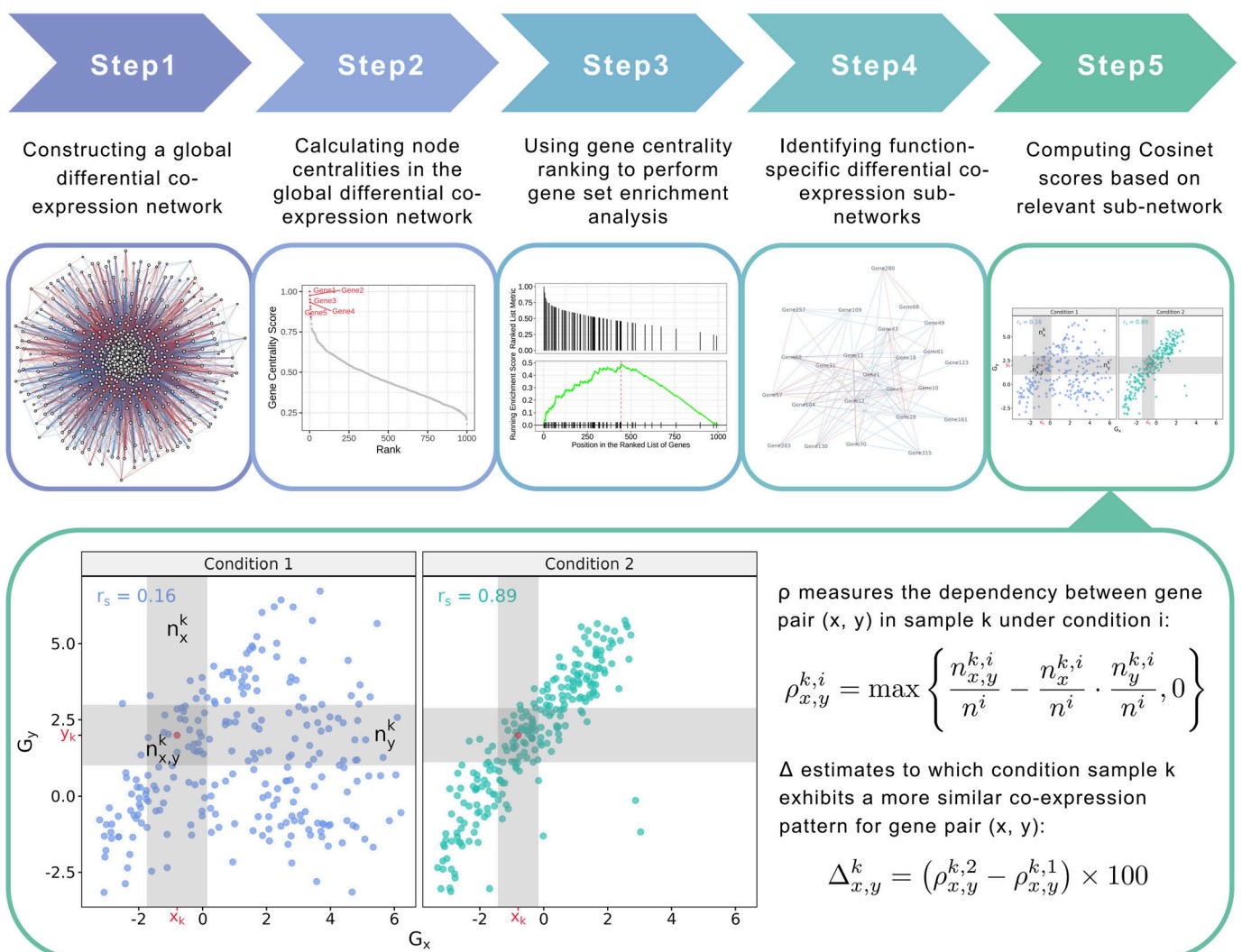

**Figure 1. Cosinet workflow.**

For network plots in step 1 and step 4, blue lines represent increased correlation coefficients, whereas red lines represent decreased correlation coefficients. In step 5, each dot on the scatter plot represents an individual sample, with the designated sample for calculating the Cosinet score shown in red. The vertical and horizontal gray regions around the red dot indicate the first and second boxes drawn around the designated sample, respectively, and their intersection represents the third box. These boxes are used for local measurement of statistical independence between two genes in the given sample.

analysis: the global network approach, the module-based approach, and the pathway-driven approach. The details of these methods are presented in Table 1.

The global network approach aims to construct a comprehensive network that includes all pairwise DCE relationships. The main feature of this approach is its ability to focus on global features of the differential network, such as connectivity, degree distribution, modularity, and entropy. The advantage of this approach is that it provides a complete view of the network and can be used to further investigate the relationship between any gene pair of interest that may be overlooked by more focused methods. It also serves as the foundation for subsequent module-based and pathway-driven analyses. However, a major drawback of this approach is its difficulty in interpretation because of the large and complex network structure. As a result, it can be challenging to focus on specific

aspects of the network without additional knowledge or analysis. Some tools that use this approach include DGCA (McKenzie et al, 2016), Discordant (Siska et al, 2016), EBcoexpress (Dawson & Kendziorski, 2012), LDGM (Tian et al, 2016), and MAGIC (Hsiao et al, 2016).

The module-based approach focuses on partitioning a network into smaller groups of co-regulated genes known as modules and identifies modules that are differentially inter- or intra-connected between two conditions. Module genes are cross-correlated with each other, which are typically defined using clustering methods. This module-based approach can be applied to identify subsystems within the network that are easier to interpret and analyze compared with a large global network. Gene ontology information can be used to test whether the identified modules are biologically meaningful. It is important to note that the results of a module-

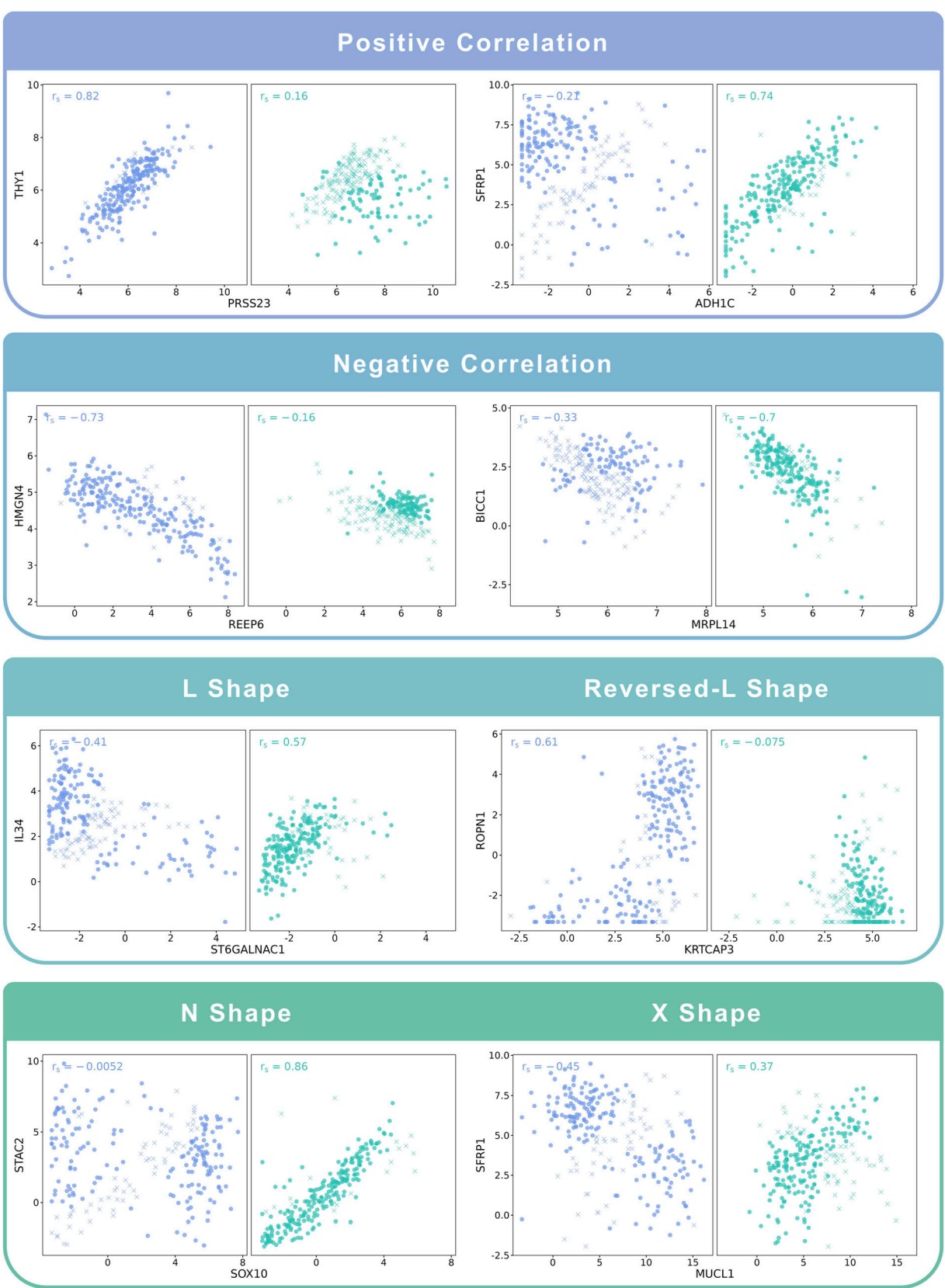

based analysis can be highly sensitive to the parameters and thresholds applied. Factors such as the choice of clustering algorithms, the number of modules to be identified, and the criteria used to define the modules can all influence the outcome of the analysis. Some of the commonly used tools for this approach include Contrast Subgraph (Lanciano et al, 2022), COSINE (Ma et al, 2011), CoXpress (Watson, 2006), DCIM (Freudenberg et al, 2010), DICER (Amar et al, 2013), DiffCoEx (Tesson et al, 2010), DiffCorr (Fukushima, 2013), and WGCNA (Langfelder & Horvath, 2008).

The pathway-driven approach detects the DCE of given gene sets that are associated with specific biological functions. This approach integrates functional analysis to reveal biologically meaningful sub-networks. The advantage of this approach lies in its ability to uncover altered co-expression relationships within a specific pathway, but it may not detect inter-pathway relationships between genes. Tools that use this approach include DysPIA (Wang et al, 2021), ESEA (Han et al, 2015), GSCA (Choi & Kendziorski, 2009), GSNCA (Rahmatallah et al, 2014), and IB-GSA (Zhang et al, 2009).

Each approach has its own merits and limitations, and the choice of approach depends on the specific research questions being addressed. In this study, we selected a pathway-driven approach for Cosinet as it offers biologically meaningful differential sub-networks for rewiring degree quantification, resulting in scores with clear biological meaning. Other tools that use this strategy, such as DysPIA (Wang et al, 2021), ESEA (Han et al, 2015), GSCA (Choi & Kendziorski, 2009), and IB-GSA (Zhang et al, 2009), compare averaged pairwise correlations or other aggregated measurements of co-expression between two conditions in a gene set between two conditions to test whether that gene set is significantly rewired. Cosinet and GSNCA (Rahmatallah et al, 2014), however, take network structure into consideration. Both Cosinet and GSNCA calculate eigenvector centralities to measure the node centrality in the network. Eigenvector centrality provides a more comprehensive picture of the network structure and highlights the most influential nodes within it. The calculation of eigenvector centrality takes into account the centrality of a node's neighbors, enabling the identification of larger regulatory networks and providing the potential to uncover disease modules within the network. The methodologies of Cosinet and GSNCA also differ from each other. GSNCA tests the null hypothesis that there is no difference in the weight vectors, which are similar to eigenvector centrality, of the genes in a gene set between two conditions by permuting the samples' condition labels and estimating the significance level. Cosinet, on the contrary, calculates the eigenvector centrality of the global DCE network and uses it as input for GSEA to identify significant pathways. GSEA provides a more nuanced evaluation of the changes in gene centrality by considering the complete shifts, rather than just comparing aggregated statistics. This permits GSEA to detect more refined patterns of pathway rewiring, such as enrichment of top rewired genes in certain pathways, instead of simply general alterations in gene centralities for a gene set. In addition, not all genes within a gene set may experience significant rewiring. The

ability to differentiate these genes is vital in reducing the noise generated by unimportant members of the gene set. In contrast to GSNCA's inability to differentiate these genes, Cosinet's workflow allows for such differentiation through the leading-edge subset analysis in GSEA.

## Association between the Cosinet score and survival outcomes in breast cancer patients

Breast cancer is the most commonly diagnosed cancer in women and the leading cause of cancer-related deaths, accounting for ~24.5% of all cancers and 15.5% of cancer-related deaths in women worldwide in 2020 (Sung et al, 2021). ~70% of breast cancers are ER+ and rely on estrogen for growth (Haque & Desai, 2019). Endocrine therapy, also known as hormone therapy, functions by inhibiting the production or effect of estrogen in the body, thereby slowing the growth of ER+ cancer cells. Currently, endocrine therapy is the mainstay of treatment for ER+ breast cancer (Gradishar et al, 2022).

We conducted a case study on a public dataset of RNA-seq expression data from breast cancer patients (Brueffer et al, 2018), to demonstrate that the Cosinet algorithm can effectively capture highly rewired pathways or functions between two conditions and accurately quantify the degree of rewiring for each individual sample (Fig 3).

We used Cosinet to construct a global DCE network between 2,832 ER+ and 241 ER− breast cancer patients. We then calculated the eigenvector centralities and performed GSEA. There are five significantly enriched gene sets, including hallmark gene sets related to E2F targets (adjusted $P = 1.1 \times 10^{-27}$, normalized enrichment score [NES]: 3.49), G2M checkpoint (adjusted $P = 9.0 \times 10^{-22}$, NES: 3.24), early estrogen response (adjusted $P = 1.5 \times 10^{-9}$, NES: 2.50), late estrogen response (adjusted $P = 1.4 \times 10^{-7}$, NES: 2.34), and mitotic spindle (adjusted $P = 3.2 \times 10^{-4}$, NES: 2.02). Previous research has shown that ER promotes an estrogen-independent, E2F-mediated transcriptional program in human breast cancer cells (Miller et al, 2011). In addition, the G2M checkpoints and mitotic spindle play direct roles in the cell cycle, which can be regulated by estrogen binding and subsequent activation of signaling pathways that promote cell cycle progression (JavanMoghadam et al, 2016; Saha et al, 2021; Zheng et al, 2023). The remaining two gene sets are also directly associated with estrogen response. All of the five enriched gene sets are known to be regulated by estrogen; thus, the GSEA step in the Cosinet workflow produced meaningful results that reflect the real biological differences between the ER− and ER+ conditions, suggesting that the quantified scores have the potential to inform personalized treatment decisions. We are interested in exploring the potential connection between Cosinet scores and clinical outcomes. To this end, we selected the gene set that defines early estrogen response, as previous research has established a substantial association between the degree of estrogen response (Oshi et al, 2020), estrogen reactivity (Takeshita et al, 2022), and percentage of ER+ cells (Morgan et al, 2011) with survival outcomes

---

**Figure 2. Examples of co-expression patterns recognized by Cosinet.**
Blue points on the left denote ER− samples, whereas green points on the right denote ER+ samples. Circles denote samples whose co-expression pattern is consistent with their annotated condition based on the sign of Cosinet scores, whereas crosses indicate inconsistent patterns.

after adjuvant endocrine therapy in breast cancer. Based on the core enriched genes of the gene set (Fig 3A), we visualized the differential sub-network (Fig 4) and used it to calculate Cosinet scores for each sample. The corresponding differential co-expression matrix for this sub-network can be found in Table S1. The distribution of the Cosinet score indicated that 93% (224/241) of ER– samples had negative scores and 85% (432/2,832) of ER+ samples had positive scores (Fig 3B), demonstrating that the co-expression patterns for most samples were consistent with their designated conditions. The Cosinet scores among ER+ samples were variable (mean ± SD: 0.61 ± 0.67; range: –2.00 to +2.25), with a considerable number of samples having scores comparable to

those of ER– samples. We hypothesized that ER+ samples with high Cosinet scores may have better survival outcomes after endocrine therapy compared to those with low scores as endocrine therapy specifically targets estrogen-responsive cells, and high Cosinet scores indicate a high level of estrogen response, whereas low scores suggest reduced responsiveness. To test this hypothesis, we constructed a Cox proportional hazard model using data from 1,603 ER+ breast cancer patients who underwent adjuvant endocrine therapy alone. The model included the Cosinet score, age, tumor size, and HER2 status (HER2+ breast cancer being a known ag-gressive subtype [Goutsouliak et al, 2020]) as covariates, with overall survival (OS) as the endpoint. The results reveal a significant

**Table 1. Comparison of differential co-expression network construction tools.**

| Type | Aim | Characteristics | Tools |
|---|---|---|---|
| Global network | To construct a comprehensive network that includes all pairwise DCE relationships | Global features of the differential network can be studied, such as connectivity, degree distribution, modularity, and entropy | DGCA (McKenzie et al, 2016), Discordant (Siska et al, 2016), EBcoexpress (Dawson and Kendziorski, 2012), LDGM (Tian et al, 2016), MAGIC (Hsiao et al, 2016) |
| Module-based | To identify groups of co-regulated genes that are differentially inter-connected under specific conditions | Focus on module genes that are cross-correlated with each other | Contrast Subgraph (Lanciano et al, 2022), COSINE (Ma et al, 2011), CoXpress (Watson, 2006), DCIM (Freudenberg et al, 2010), DICER (Amar et al, 2013), DiffCoEx (Tesson et al, 2010), DiffCorr (Fukushima, 2013), WGCNA (Langfelder and Horvath, 2008) |
| Pathway-driven | To measure DCE of given gene sets that are related to certain biological functions | Combined with functional analysis to produce highly rewired sub-networks that are biologically meaningful | Cosinet, DysPIA (Wang et al, 2021), ESEA (Han et al, 2015), GSCA (Choi and Kendziorski, 2009), GSNCA (Rahmatallah et al, 2014), IB-GSA (Zhang et al, 2009) |

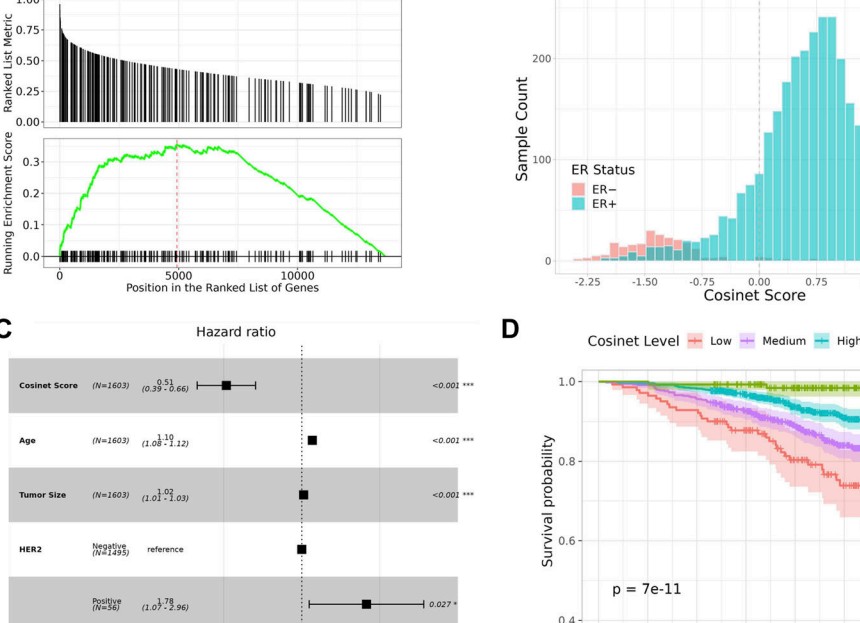

**Figure 3. Higher Cosinet scores are associated with a better prognosis for breast cancer patients receiving adjuvant endocrine therapy.**
**(A)** Highly rewired genes between 2,832 ER+ and 241 ER– breast cancer patients are enriched for genes that define early estrogen response. Genes located before the red line represent core enrichment genes.
**(B)** Distribution of Cosinet scores calculated on the basis of the differential co-expression network regarding early estrogen response. **(C)** Multivariate Cox regression analysis of 1,603 ER+ breast cancer patients treated with adjuvant endocrine therapy alone, with Cosinet scores, age at diagnosis, tumor size, and HER2 status as covariates, and overall survival (OS) as the endpoint. Hazard ratios and 95% confidence ranges are shown. **(D)** Kaplan–Meier plot for 1,603 ER+ breast cancer patients treated with adjuvant endocrine therapy alone, stratified by Cosinet scores (low: <0; medium: 0–0.75; high: 0.75–1.5; very high: ≥1.5), with OS as the endpoint. Shaded bands represent the confidence intervals.

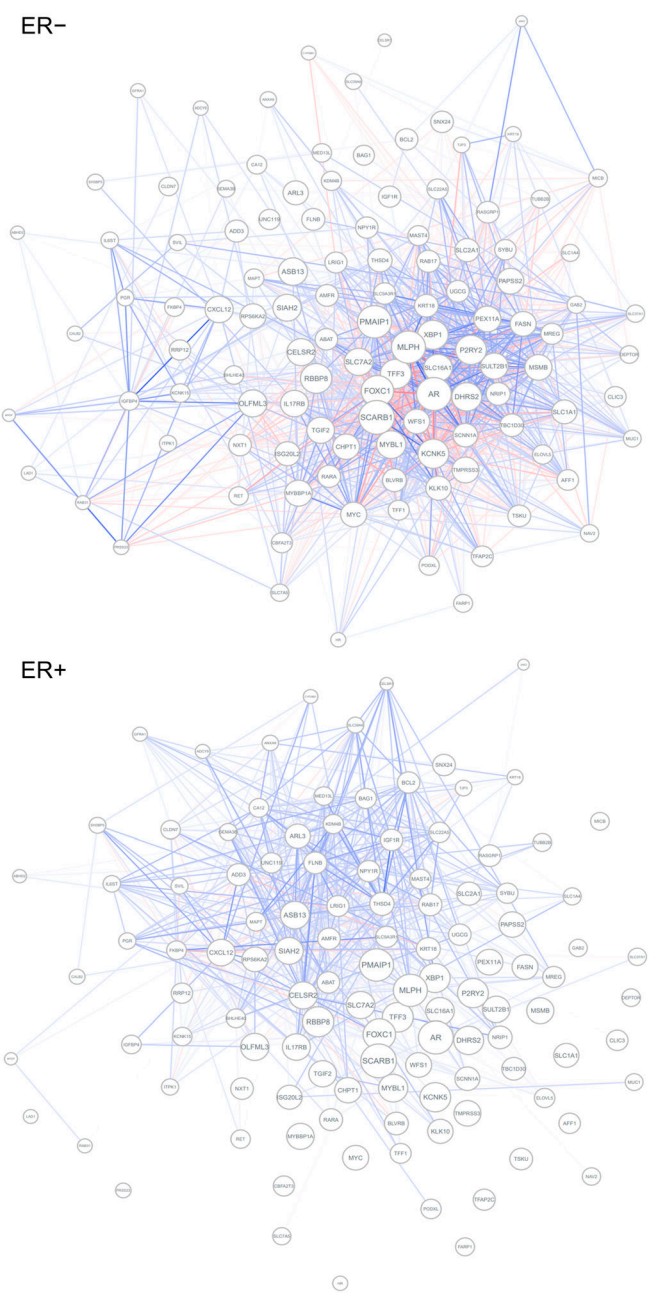

**Figure 4. Differential co-expression network of ER+ samples and ER− samples regarding early estrogen response.**
Blue edges represent positive correlations, whereas red edges represent negative correlations. Edge width and opacity indicate the strength of correlation. Edges for gene pairs with an absolute correlation coefficient less than 0.3 are hidden.

relationship between the Cosinet score and the hazard of death ($P$ = $4.12 \times 10^{-07}$, hazard ratio: 0.51), as demonstrated in Fig 3C. Specifically, holding the other covariates constant, an increase of one score in the Cosinet score reduces the hazard of death by 49% (95% CI: 33.7–60.5%). Thus, we conclude that a higher Cosinet score is associated with a better prognosis. To directly visualize the overall effect of Cosinet scores on prognosis, we categorized them into four levels: low, medium, high, and very high, and plotted

the corresponding survival curves. A cutoff value of 0 was used to distinguish samples with low Cosinet levels from the others, as scores below 0 indicate greater similarity to ER− samples. The other two cutoffs, 0.75 and 1.5, were determined based on the score distribution. Fig 3D shows the Kaplan–Meier plot for the 1,603 ER+ breast cancer patients treated with adjuvant endocrine therapy alone. The 5-yr Kaplan–Meier survival rates for patients with low, medium, high, and very high Cosinet levels were 73.9% (95% CI: 65.9–82.8%, n = 144), 84.0% (95% CI: 80.7–87.4%, n = 588), 90.9% (95% CI: 88.5–93.4%, n = 732), and 98.4% (95% CI: 96.2–100.0%, n = 142), respectively ($P = 7 \times 10^{-11}$, log-rank test).

**Validation of Cosinet scores in a separate dataset**

To validate our findings, we used the DCE sub-network structure identified in the previous dataset as the basis for calculating Cosinet scores for a separate dataset of 3,516 breast cancer patients (Staaf et al, 2022) (Fig 5). To simplify terminology, we refer to the first dataset as the primary dataset and the second dataset as the validation dataset. The distribution of Cosinet scores in the validation dataset closely resembled that of the primary dataset, with 95% (463/488) of the ER− samples having a negative Cosinet score and 85% (2,586/3,028) of the ER+ samples having a positive Cosinet score (Fig S2). These results suggest that the differences in estrogen response quantified by our Cosinet algorithm reflect real biological differences that are consistent across datasets.

Next, we evaluated whether the association between Cosinet scores and clinical outcomes remained valid. The validation dataset contained data on OS, as well as recurrence-free interval, distant recurrence–free interval (DRFi), and breast cancer–free interval (BCFi). We performed Cox proportional hazard regression analyses separately for each of these four endpoints. The model included the same covariates as used in the primary dataset, which are the Cosinet score, age, tumor size, and HER2 status. The results once again demonstrated a strong association between Cosinet scores and clinical outcomes (Fig 5). Specifically, the multivariable hazard ratios of the Cosinet score were 0.33 (95% CI: 0.21–0.52, $P = 1.71 \times 10^{-6}$) for RFi, 0.42 (95% CI: 0.24–0.75, $P$ = 0.0036) for DRFi, 0.33 (95% CI: 0.16–0.69, $P$ = 0.0033) for BCFi, and 0.69 (95% CI: 0.51–0.92, $P$ = 0.013) for OS. To directly visualize the overall effect of Cosinet scores on prognosis, Fig S3 shows the Kaplan–Meier plots for the validation dataset, using RFi, DRFi, BCFi, and OS as endpoints. We used identical cutoffs for categorization as those applied in the primary dataset. The results show that the original cutoffs continue to have a significant discriminatory effect in the validation dataset.

In the context of module-based co-expression analysis, the eigengene serves as a commonly used representative expression profile of the gene module. It is computed for each gene module by deriving the first principal component from the expression profiles of all genes contained within that module. This approach facilitates the exploration of associations between modules and external variables, such as disease status or treatment response. However, it is important to note that unlike Cosinet, which quantifies variation in co-expression patterns, the eigengene essentially functions as a

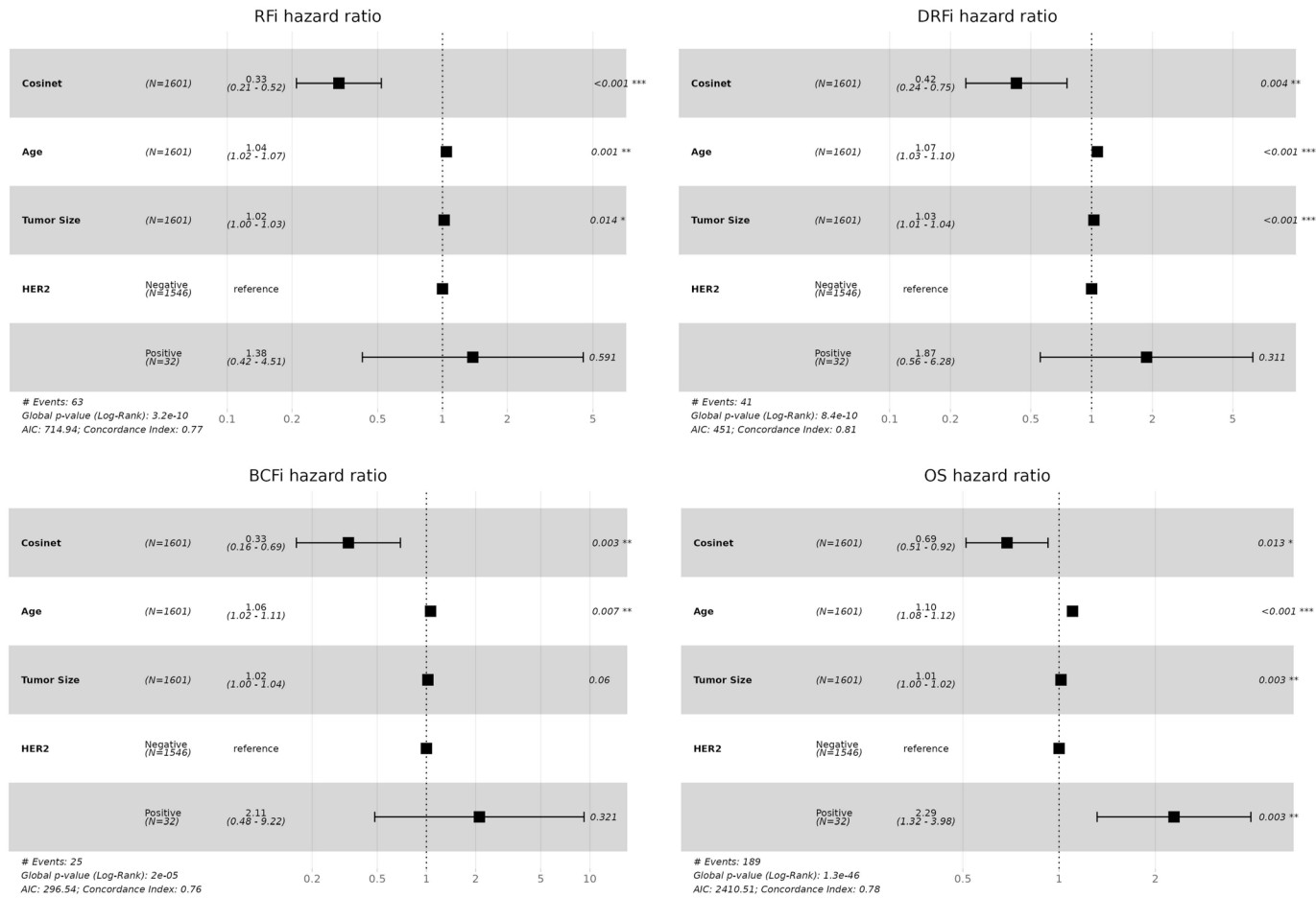

**Figure 5. Validation of the association between Cosinet scores and clinical outcomes in a separate dataset.**
The figure shows the results of multivariate Cox regression analysis performed on 1,601 ER+ breast cancer patients from the validation dataset who received adjuvant endocrine therapy alone. The differential co-expression sub-network structure identified in the primary dataset was used as the basis for calculating Cosinet scores. The analysis included Cosinet scores, age at diagnosis, tumor size, and HER2 status as covariates. Recurrence-free interval, distant recurrence–free interval, breast cancer–free interval, and overall survival (OS) were evaluated as endpoints. Hazard ratios with their corresponding 95% confidence intervals are displayed.

weighted average expression profile that captures the predominant expression pattern within the module. It provides a summary of the primary variation in gene expression observed within the module across different samples. Consequently, Cosinet provides a measure at the level of gene–gene co-expression relationship that is not present in the eigengene. We further evaluated the predictive performance of Cosinet and eigengenes for 5-yr clinical outcomes in both datasets. Specifically, we identified a single module for ER+ samples and a consensus module for both ER+ and ER– samples based on the selected estrogen response gene set using WGCNA (Langfelder & Horvath, 2008), and then calculated the corresponding module eigengenes. Threefold cross-validation was performed on each dataset, and the resulting area under the receiver operating characteristic curve (AUC) for the test sets is shown in Fig S4 (with detailed AUC values provided in Table S2). Our results show that Cosinet outperformed eigengenes in all tasks, with an overall improvement of 0.09 over a single module eigengene and 0.1 over a consensus module eigengene.

Staaf et al (2022) developed a single-sample predictive model for risk of recurrence (SSP-ROR) based on this dataset, which uses machine-learning approaches to train a set of decision rules of the form "If the expression of gene A is less than the expression of gene B, then we tend to classify the patient as subtype X." and integrates the resulting rules via a single naive Bayes classifier. To assess the significance of our Cosinet score after accounting for the SSP-ROR risk category, we included it in our multivariable Cox proportional hazard model. In line with the analysis of Staaf et al (2022), we also included lymph node status and Nottingham histologic grade as covariates. The resulting multivariable hazard ratios of the Cosinet score were similar to those before: 0.31 (95% CI: 0.19–0.50, $P = 2.28 \times 10^{-6}$) for RFi, 0.44 (95% CI: 0. 23–0.84, $P = 0.013$) for DRFi, 0.32 (95% CI: 0.15–0.68, $P = 0.0028$) for BCFi, and 0.68 (95% CI: 0.49–0.94, $P = 0.018$) for OS as endpoints, as shown in Fig S5. In comparison, other factors showed insignificant or weaker associations. For example, the SSP-ROR was only significant for BCFi, suggesting that the Cosinet scores provided additional information beyond what was trained and captured by the SSP-ROR model. Overall, the results from the validation set confirmed the validity of our previous findings.

# Discussion

The results of this study demonstrate that Cosinet is a valuable and robust tool for DCE analysis at the single-sample level, with the potential to facilitate personalized treatment decisions. By accurately quantifying the degree of rewiring in individual samples compared with reference conditions using mechanism-related DCE sub-networks, Cosinet provides early insights into the potential efficacy of treatment without relying on survival data. This represents a significant advantage over other transcriptome-based approaches that require survival data for risk prediction (Parker et al, 2009; Zhang et al, 2020; Győrffy, 2021). Our tool is useful for selecting patients who are most likely to benefit from a certain treatment with known or particularly novel targets when the mechanism of action can be traced through the co-expression network. Based on the results of Cosinet and other relevant clinical factors, alternative treatments or combination therapies can be considered for patients who are less suitable for the treatment being evaluated, ultimately leading to improved treatment outcomes. In our case study, patients with low-to-medium Cosinet scores exhibited a lower activity of estrogen response, suggesting that oncologists may consider switching to or combining with drugs that target other pathways, such as the mTOR inhibitor everolimus (Baselga et al, 2012) or the PI3Kα inhibitor alpelisib (André et al, 2019). The power of Cosinet in identifying patients with highly favorable outcomes to endocrine therapy may also avoid unnecessary trials of other treatments on these patients, allowing patients to avoid unnecessary side effects and toxicities, such as hyperglycemia, rash, and diarrhea.

Molecular subtypes of breast cancer have been well defined by the PAM50 gene signature that is based on the expression pattern of 50 genes (Parker et al, 2009). These PAM50 subtypes are known to be associated with distinct clinical outcomes and treatment responses (Wallden et al, 2015). We have also included the PAM50 subtype in the multivariable Cox regression model. The result suggests that after adjusting for Cosinet score, age, tumor size, and HER2 status, the significance of PAM50 subtypes is only marginally evident, whereas the significance of all other covariates remains robust, yielding hazard ratios similar to those obtained in the model without PAM50 (Fig S6). In addition, we plotted Kaplan–Meier survival curves stratified by PAM50 subtypes using the data from 1,603 ER+ patients who underwent adjuvant endocrine therapy alone (Fig S7). In comparison with the PAM50 subtyping approach, our categorization of Cosinet scores shows a clearer and more distinct separation between different groups (Fig 3D). Notably, our categorization identifies a "very high" group characterized by an exceptionally high survival rate (5-yr survival rate: 98.4%), which is not captured by the PAM50 classification. Furthermore, we observed that only ~38 samples with low expression of the ER-coding gene (ESR1) and low Cosinet scores show a positive correlation between the ESR1 expression and Cosinet scores (Fig S8). This indicates that the low expression of ESR1, which is known to contribute to endocrine therapy resistance in ER+ breast cancer (Kim et al, 2011), only accounts for few samples displaying low estrogen response activity as detected by Cosinet, and high Cosinet scores do not indicate high ESR1 expression. Apart from the ESR1 expression level,

the diverse estrogen response reactivity of ER+ samples captured by Cosinet may be attributed to disease heterogeneity, genetic variation, differences in the activity of up- and downstream signaling pathways, or other factors that can affect estrogen response levels. The categorization of the Cosinet scores allowed for simplified visualization of the relationship with survival outcomes. Although this approach facilitated the interpretation of the results, it is important to recognize the trade-off in information loss because of the categorization process. By categorizing the Cosinet scores, we aimed to provide a clear representation of the overall trends observed in the data. However, it is worth noting that using the original continuous scores would retain the full granularity and allow for a more nuanced understanding of the relationship between Cosinet scores and prognosis.

Differential expression can have an impact on DCE, from both a biological and a mathematical perspective, thereby influencing quantification. However, as shown in Fig 2, many rewired genes do not exhibit differential expression, rendering them undetectable with the traditional single-gene-wise approaches. To further illustrate this point, we generated a heatmap (Fig S9) demonstrating the absence of a discernible pattern between DCE network gene expression and Cosinet scores, indicating that Cosinet scores are not solely indicative of variances in gene expression levels but rather reflecting the co-expression relationships. Previously, Oshi et al used the degree of early estrogen response to predict survival after endocrine therapy (Oshi et al, 2020). They employed the GSVA algorithm (Hänzelmann et al, 2013), which evaluates whether a gene is highly or lowly expressed in a given sample within the sample population distribution, and incorporates the GSEA approach to generate sample-wise gene set enrichment scores. This method relies on the relative gene expression level to assess the activity level of a given gene set for a single sample. In contrast, our approach extracts information in a network-based manner and quantifies differences in gene–gene relationships between two conditions.

Cosinet offers a high degree of flexibility in its workflow, allowing researchers to incorporate intermediate results from other tools. For example, one can use different methods for generating DCE matrices, such as mutual information–based approaches, adopt alternative gene centrality ranking methods, or use predefined sub-network structures that align with their research problem to calculate Cosinet scores. This adaptability allows the calculation of Cosinet scores that can be tailored to a specific research question. Moreover, Cosinet can be applied to a wide range of research scenarios and is not limited to specific disease types or treatments.

# Materials and Methods

### Constructing a global DCE network

To construct a global DCE network, we first used the gene expression data to calculate Spearman's correlation coefficient matrix for each condition. Spearman's correlation coefficient, denoted as $r_s$, is a nonparametric measure of the correlation between the rankings of two variables and is defined as follows:

$$r_s = \frac{\text{cov}(R(X), R(Y))}{\sigma_{R(X)}\sigma_{R(Y)}} \tag{1}$$

where R(X) and R(Y) represent the rank order of gene expression for genes X and Y, cov(R(X), R(Y)) is the covariance between the rank variables R(X) and R(Y), and $\sigma_{R(X)}$ and $\sigma_{R(Y)}$ are the standard deviations of R(X) and R(Y), respectively. We chose this method because it is tolerant to the presence of outliers and does not require a linear relationship between the variables being analyzed. Gene expression data may not follow a linear relationship, as the relationship between the expression levels of different genes can be complex and influenced by various factors.

Next, we used Fisher's z-transformation to compare the correlation coefficients of a gene pair under two conditions. First, we applied Fisher's transformation to convert the correlation coefficient into a normally distributed variable, which makes it easier to perform statistical tests. The transformation is given by

$$z_r = \frac{1}{2}\ln\left(\frac{1 + r_s}{1 - r_s}\right) \tag{2}$$

We then used the z test to determine the significance of the difference between the two transformed correlation coefficients. It is calculated with the following formula (Sheskin, 2011):

$$Z = \frac{z_{r,2} - z_{r,1}}{\sqrt{\frac{c}{n_2 - 3} + \frac{c}{n_1 - 3}}} \tag{3}$$

where $z_{r,1}$ and $z_{r,2}$ are the Fisher-transformed values for the two correlation coefficients being compared, $n_1$ and $n_2$ are the sample sizes for the two groups being compared, and c is a constant value that depends on the type of correlation coefficient. For Pearson's correlation coefficients, c = 1, and for Spearman's rank correlation coefficients, c = 1.06 (Sheskin, 2011). This z-score–based method pools the variances of the transformed coefficients, resulting in an improved error estimate for the difference statistic (Bhuva et al, 2019). It has been shown by Bhuva et al (2019) to be the best performer for detecting conditional relationships in gene expression data.

We removed genes with expression values greater than zero in fewer than 20 samples in any condition to ensure that the calculated correlations were based on a sufficient amount of data. For the remaining genes, we calculated the z-score for each gene pair. The resulting matrix, denoted by Z, represents the global DCE network and demonstrates the strength of correlation changes between the two conditions.

## Calculating node centralities in a DCE network

The centrality ranking of nodes in a DCE network identifies important genes that may be involved in various biological processes and play key roles in regulating gene expression. We used eigenvector centrality to measure the node centrality in the network, which is an effective method for identifying influential nodes in a network. Eigenvector centrality is used to measure the importance of a node by considering the centrality of its neighbors. Specifically, the eigenvector centrality of a node is proportional to the sum of the eigenvector centralities of its neighbors. This means that the eigenvector centrality of a node with many connections to high-centrality nodes is high, whereas a node with few connections to high-centrality nodes is low. If a particular gene is identified as highly central in the network, it may be important in regulating the expression of several other genes. However, if that gene is also connected to other highly central genes, it may be part of a larger regulatory network that is involved in a particular biological process or pathway. By considering the centrality of a node's neighbors, it is possible to uncover larger regulatory networks and identify potential disease genes or disease modules within the network. This can be useful for ranking genes in DCE analysis, as it allows for a more comprehensive assessment of the importance of individual genes in the context of the overall network.

To calculate the eigenvector centrality scores for the nodes in the global DCE network, we used the power iteration Algorithm 1:

Algorithm 1 Power Iteration to Calculate Eigenvector Centrality.

1: **procedure** EigenvectorCentrality (Z, tolerance, maxIter)
2: M ← |Z| ▷ M is the absolute value of the matrix Z
3: n ← size(M) ▷ n is the size of the square matrix M
4: v ← $0^n$ ▷ $0^n$ is a vector of 0's with length n
5: v' ← $\frac{1^n}{n}$ ▷ $\frac{1^n}{n}$ is a vector of $\frac{1}{n}$ s with length n
6: iteration ← 0
7: **while:** |v − v' | > tolerance and iteration < maxIter **do**
8: v ← v'
9: v' ← Mv'
10: v' ← $\frac{v'}{\max(v')}$ ▷ scale v' to have a maximum value of 1
11: iteration ← iteration + 1
12: **end while**
13: **return** v'
14: **end procedure**

## Identifying function-specific DCE sub-networks

To uncover biological processes and pathways that may be differentially regulated between two conditions, we conducted GSEA (Subramanian et al, 2005) using gene centrality ranking in the DCE network. This allows us to identify gene sets whose members are overrepresented at the top of the ranked list, suggesting enrichment for a specific function or pathway. It is important to note that gene sets can be defined using various methods, and not all members of a gene set necessarily participate in a biological process (Subramanian et al, 2005). To identify the "core" members of a gene set that contribute to the enrichment signal, Subramanian et al defined the "leading-edge subset" as the genes in the gene set that appear in the ranked list at or before the point where the running sum for the enrichment score reaches its maximum deviation from zero (Subramanian et al, 2005). The leading-edge subset, also referred to as the core enrichment genes of a significantly enriched gene set, is denoted as G. G, along with gene pairs in G that show changes in gene–gene association between two conditions, forms a function-specific DCE sub-network. Researchers can focus on sub-networks that are relevant to their research questions, such as disease processes or other specific areas of

interest, by examining the functional annotations of the enriched gene sets. Through this process, we can gain valuable insights into the large and complex DCE network and quantify the degree of rewiring in DCE sub-networks relevant to specific functions for individual samples. Downstream analysis can then be performed to further investigate these differences.

In the GSEA that we performed, we used the Molecular Signatures Database (MSigDB) hallmark gene sets (Subramanian et al, 2005) from the msigdbr R package (version 7.5.1; https://cran.r-project.org/package=msigdbr). MSigDB is a comprehensive collection of annotated gene sets that are widely used for GSEA. The hallmark gene sets in MSigDB are a collection of gene sets that represent the most important and well-established biological pathways and processes (Liberzon et al, 2015). We performed the GSEA using clusterProfiler (version 4.4.4) (Wu et al, 2021) and set the cutoff for Hochberg's adjusted P-value to 0.01.

## Computing Cosinet scores based on DCE sub-network

Next, we used the DCE sub-networks to calculate Cosinet scores for individual samples. The Cosinet score reflects the degree to which the gene co-expression patterns of a sample in a differential sub-network resemble those of the reference conditions. There are three steps to calculate the Cosinet score: (1) estimate the statistical independence of a gene pair in a given sample using the statistic $\rho$; (2) compare the sample's gene co-expression pattern with the reference conditions by subtracting the statistic $\rho$ of the two conditions; and (3) aggregate the gene pair comparisons at the sub-network level to produce the final Cosinet score.

The first step was inspired by the CSN method introduced by Dai et al (2019), which is a method for determining the association between genes at the single-cell level using single-cell RNA-sequencing (scRNA-seq) data. The authors derived a local measurement of statistical independence from probability theory and used it to define a statistic, $\rho$, that measures the independence of two genes within a given cell. To adapt this method for our purposes, we modified the statistic to measure the dependency of a gene pair (x,y) for a single sample k in two conditions i = 1, 2 using bulk RNA-seq data. To do this, we first calculated the $\rho$ statistic, which is based on the frequencies of cells in three boxes drawn around the designated cell in a scatter plot of the gene expression values. The first box encompasses the range of gene x expression values near $x_k$, the second box encompasses the range of gene y expression values near $y_k$, and the third box is the intersection of the first and second boxes (see Fig 1). The variables $n_x^{k,i}$, $n_y^{k,i}$, and $n_{x,y}^{k,i}$ represent the number of samples in the first, second, and third boxes in condition i, respectively, and $n^i$ represents the number of samples under condition i. The formula for estimating the dependency of gene pair (x,y) in sample k for the two conditions is given by

$$\rho_{x,y}^{k,i} = \max\left\{ \frac{n_{x,y}^{k,i}}{n^i} - \frac{n_x^{k,i}}{n^i} \cdot \frac{n_y^{k,i}}{n^i}, 0 \right\} \tag{4}$$

If genes x and y are independent in sample k, then the value of $\rho_{x,y}^k$ is equal to 0. If the genes are dependent, then $\rho_{x,y}^k$ is greater than 0. If $\rho_{x,y}^k$ is less than 0, it suggests that sample k is an outlier in the co-expression plot. In this case, we set $\rho_{x,y}^k$ to 0 and consider the genes to be independent in sample k. Specifically, we plotted the designated sample on the scatter plot for each condition and calculated the two P-values, regardless of whether it belongs to either of the two conditions or remains unclassified. In the original CSN method, the statistic was further normalized based on the assumption that most gene pairs are independent of each other to set a threshold for the single-cell network. We did not follow this step, as we are not constructing individual networks for single samples, but rather aggregating gene pairs at the network level for quantification. Furthermore, the gene pairs in our sub-networks are highly correlated, which means that the independence assumption on which the normalization is based is not satisfied. For a more robust comparison between the two conditions, we modified the algorithm to draw boxes around points in the scatter plot using a predefined range of expression values, rather than a predefined number of neighbors as in the original CSN method (Dai et al, 2019). This range is defined as a certain proportion b of the expression range, with a default value of b = 0.1 (which corresponds to 10% of the difference between the maximum and minimum expression values, equally distributed on either side of the sample point). Using a fixed box size allows for more consistent measurements when the local densities between the two conditions differ greatly in a certain region. To mitigate the impact of extreme expression values on box size determination, users can specify the percentage of intermediate quantiles (with a default of 1, the full expression range) of expression values to be used as the basis for box size calculation, effectively excluding the most extreme values.

To determine to which condition the sample exhibits a more similar gene co-expression pattern, we calculated the subtraction of the statistic $\rho$, denoted as $\Delta$:

$$\Delta_{x,y}^k = \left( \rho_{x,y}^{k,2} - \rho_{x,y}^{k,1} \right) \times 100 \tag{5}$$

Note that we multiply the resulting decimal by 100 to convert it to a percentage.

To calculate the final Cosinet score for a single sample, we took the average of the $\Delta$ values for gene pairs that pass a threshold for the z-score, weighted by the absolute value of their z-scores. This is given by the following equation:

$$CS(k) = \frac{1}{|G_{\geq t}^2|} \sum_{(x,y) \in G_{\geq t}^2} |z_{x,y}| \Delta_{x,y}^k \tag{6}$$

where CS(k) is the Cosinet score of sample k, $G_{\geq t}^2$ is the subset of $G^2$ that consists of gene pairs (x,y) with $|z_{x,y}| \geq t$, and t is a threshold that filters out gene pairs with little change in co-expression patterns. The final Cosinet score for each sample represents a measure of how closely the gene co-expression patterns of that sample match the patterns observed in the reference conditions within the differential sub-network. A lower Cosinet score indicates that the gene co-expression patterns of the given sample are more similar to those of condition 1, whereas a higher Cosinet score indicates that they are more similar to those of condition 2.

## Survival analysis

All analyses were conducted in R (version 4.2.0). To analyze survival outcomes, we used the survival R package (version 3.4.0; https://CRAN.R-project.org/package=survival). We employed the Kaplan–Meier method to estimate survival curves and compared them using the log-rank test. Hazard ratios were calculated by the multivariable Cox regression model using the R function coxph. The assumption of proportional hazards was met based on the analysis of Schoenfeld residuals using the survival package. We used the survAUC R package (version 1.2.0; https://cran.r-project.org/package=survAUC) to compute the AUC for the survival outcome.

# Data Availability

For this study, we used two publicly available breast cancer RNA-seq datasets sourced from two publications (Brueffer et al, 2018; Staaf et al, 2022). We chose these two datasets because of their large sample sizes, recent data (published in 2018 and 2022, respectively), complete and detailed clinical annotations, and consistent data collection, which together provide a strong foundation for comprehensive and up-to-date research on ER+ breast cancer outcomes with hormone therapy. The first dataset (primary dataset) contains data from 3,073 breast cancer patients (Brueffer et al, 2018) (https://www.ncbi.nlm.nih.gov/geo/query/acc.cgi?acc=GSE96058), whereas the second dataset (validation dataset) contains data from 3,516 breast cancer patients (Staaf et al, 2022; Vallon-Christersson, 2023) after removing overlapping samples to the first dataset. Both datasets contain $\log_2$(FPKM + 0.1) values (FPKM stands for Fragments Per Kilobase of transcript per Million mapped reads) of gene expression. The Cosinet R package is publicly available on GitHub (https://github.com/LanyingWei/cosinet) with detailed step-by-step instructions and examples.

# Supplementary Information

# Acknowledgements

We thank Dr. G Peng, Dr. W Zheng, and Dr. K Tian from the Innovation Center for their helpful discussions. We also thank Y Wang, S Guo, X Xiang, J Zhou, and all members of the Beijing StoneWise Technology Co Ltd. for administrative support.

## Author Contributions

L Wei: conceptualization, resources, data curation, software, formal analysis, validation, investigation, visualization, methodology, and writing—original draft.
Y Xin: investigation and writing—review and editing.
M Pu: investigation and writing—review and editing.
Y Zhang: conceptualization, supervision, funding acquisition, project administration, and writing—review and editing.

## Conflict of Interest Statement

L Wei, Y Xin, M Pu, and Y Zhang are employees of StoneWise.

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
