## [Reviewer comments · Life Science Alliance]

Life Science Alliance

Patient-Specific Analysis of Co-expression to Measure Biological Network Rewiring in Individuals

Lanying Wei, Yucui Xin, Mengchen Pu, and Yingsheng Zhang

DOI: <https://doi.org/10.26508/lsa.202302253>

Corresponding author(s): Yingsheng Zhang, Beijing StoneWise Technology Co Ltd and Lanying Wei,

Review Timeline:

Submission Date:	2023-07-06
Editorial Decision:	2023-08-21
Revision Received:	2023-09-29
Editorial Decision:	2023-11-03
Revision Received:	2023-11-04
Accepted:	2023-11-06

Transaction Report:

August 21, 2023

Re: Life Science Alliance manuscript #LSA-2023-02253-T

Yingsheng Zhang
Beijing StoneWise Technology Co Ltd

Dear Dr. Zhang,

Thank you for submitting your manuscript entitled "Patient-Specific Analysis of Co-expression Network to Quantify the Rewiring Degree of Biological Network in Individuals" to Life Science Alliance. The manuscript was assessed by expert reviewers, whose comments are appended to this letter. We invite you to submit a revised manuscript addressing the Reviewer comments.

Thank you for this interesting contribution to Life Science Alliance. We are looking forward to receiving your revised manuscript.

Sincerely,

B. MANUSCRIPT ORGANIZATION AND FORMATTING:

Reviewer #1 (Comments to the Authors (Required)):

This manuscript describes a novel bioinformatics tool that helps measure how different biological networks are in individuals. It focuses on studying the different biological processes related to two states, such as different disease subtypes. The authors developed a method to estimate how closely or differently a sample represents its assigned state based on specific biological processes of interest. The quantified scores hold the potential to predict treatment outcomes, particularly for treatments targeting those specific processes. The authors provide supporting evidence by analyzing ER+ and ER- breast cancer samples in two datasets. The results show that a closer match between the estrogen response network in individual ER+ samples and that of general ER-positive samples is associated with better treatment outcomes after hormone therapy.

Overall, the tool shows potential as a new approach to characterize individual samples in relation to the disease process, which may have practical value in predicting clinical outcomes. Nevertheless, there are some points that could be further improved:

1. The code should be uploaded to public platform. Additionally, providing detailed tutorial and example use cases would be beneficial for potential users to better understand and follow the algorithm, since there are multiple steps.
2. Regarding the algorithm, the authors mentioned that the box region length is a certain proportion of the expression region, which is the difference between the maximum and minimum expression values. But what if there are outliers so that the maximum or minimum values are too extreme? Consider modifying the algorithm so that it is more tolerant to outliers.
3. In the validation section, the authors present hazard ratio plots, but it would be valuable to include survival curves of categorized score levels as well. Also, does the categorization criteria in the first data set apply well to the validation data set?
4. Has the method been tested for its applicability to single-cell RNA-seq data?
5. The genes and the corresponding matrix representing the early estrogen response sub-network that was used in the analysis should be provided to offer transparency and reproducibility.
6. Consider including the enrichment scores for GSEA results.
7. For the GSEA plot, I recommend highlighting the core enrichment part to help readers better understand what "core enrichment" means.

Reviewer #2 (Comments to the Authors (Required)):

Wei et al define a single-sample differential co-expression method combining the previously described z-score based method for differential co-expression and the rho statistics (Dai et al, 2019) for quantifying co-expression in single cells. Moreover, they add a pathway-enrichment step to select genes with high levels of network rewiring within a highly-rewired pathway. To assess the reliability of their method, they apply it to two breast cancer datasets, comparing ER+ and ER- tumors and testing the relationship between their differential co-expression score and patients' outcome upon endocrine therapy. The manuscript is clear and well written, and the topic is highly relevant to the field.

Major comments:

- Alternative measures of co-expression modules in single samples exist (e.g. module eigengene as defined in WGCNA). How well do they perform in predicting patients' outcome upon endocrine therapy, when based on the selected estrogen response gene set? The authors should show the results using the module eigengene and comment on the differences with their metric.
- The code for Cosinet should be made available in a public repository, ideally as an R package

Minor comments:

- Methods for quantifying co-expression networks in single samples have been developed (e.g. <https://doi.org/10.1093/bib/bbaa268> and reviewed in <https://doi.org/10.1093/bib/bbz089>) and should be mentioned in the introduction.
- In the list of module-based DCE methods, the method developed by Lanciano et al., 2022 is missing (doi: 10.1093/gigascience/giad010).
- Figure 3b is not readable.
- All R packages employed should be cited
- The choice of the two breast cancer datasets (amongst the hundreds that have been published) should be motivated.

We thank the reviewers for their insightful comments and suggestions on our manuscript. We have carefully addressed each of these points as follows:

Reviewer #1 (Comments to the Authors (Required)):

This manuscript describes a novel bioinformatics tool that helps measure how different biological networks are in individuals. It focuses on studying the different biological processes related to two states, such as different disease subtypes. The authors developed a method to estimate how closely or differently a sample represents its assigned state based on specific biological processes of interest. The quantified scores hold the potential to predict treatment outcomes, particularly for treatments targeting those specific processes. The authors provide supporting evidence by analyzing ER+ and ER- breast cancer samples in two datasets. The results show that a closer match between the estrogen response network in individual ER+ samples and that of general ER-positive samples is associated with better treatment outcomes after hormone therapy.

Overall, the tool shows potential as a new approach to characterize individual samples in relation to the disease process, which may have practical value in predicting clinical outcomes. Nevertheless, there are some points that could be further improved:

We thank the reviewer for the valuable feedback and recognition of our work.

1.The code should be uploaded to public platform. Additionally, providing detailed tutorial and example use cases would be beneficial for potential users to better understand and follow the algorithm, since there are multiple steps.

We have added in the Data Access section: "The Cosinet R package is publicly available on GitHub (<https://github.com/LanyingWei/cosinet>) with detailed step-by-step instructions and examples".

2.Regarding the algorithm, the authors mentioned that the box region length is a certain proportion of the expression region, which is the difference between the maximum and minimum expression values. But what if there are outliers so that the maximum or minimum values are too extreme? Consider modifying the algorithm so that it is more tolerant to outliers.

We thank the reviewer for pointing this out. We have modified the algorithm by adding a parameter called "binRange" to address this potential issue. In the Methods section, we have added: "To mitigate the impact of extreme expression values on box size determination, users can specify the percentage of intermediate quantiles (with a default of 1, the full expression range) of expression values to be used as the basis for box size calculation, effectively excluding the most extreme values". In addition, the Pearson correlation coefficients of the Cosinet scores between using the full expression range and using the range ignoring the most extreme 1%, 3% and 5% expression values are 0.98, 0.95 and 0.94 in our case, indicating that in our analysis the Cosinet scores are generally not influenced by the possible outliers.

3.In the validation section, the authors present hazard ratio plots, but it would be valuable to

include survival curves of categorized score levels as well. Also, does the categorization criteria in the first data set apply well to the validation data set?

In section 2.5, we added: "To directly visualize the overall effect of Cosinet scores on prognosis, Fig. S3 shows the Kaplan-Meier plots for the validation dataset, using RFi, DRFi, BCFi and OS as endpoints. We used identical cutoffs for categorization as those applied in the primary dataset. The results show that the original cutoffs continue to have a significant discriminatory effect in the validation dataset".

4. Has the method been tested for its applicability to single-cell RNA-seq data?

No, the method has not been tested for its applicability to single-cell RNA-seq data. While we recognize the importance of exploring its potential usefulness in this context, it is currently an avenue for future research. Future work may involve adapting the method to accommodate the unique characteristics and challenges presented by single-cell RNA-seq data and conducting rigorous testing and validation in single-cell datasets to assess its performance.

5. The genes and the corresponding matrix representing the early estrogen response sub-network that was used in the analysis should be provided to offer transparency and reproducibility.

In section 2.4, we added: "The corresponding differential co-expression matrix for this sub-network can be found in Table S1".

6. Consider including the enrichment scores for GSEA results.

We have included the normalized enrichment scores in the text.

7. For the GSEA plot, I recommend highlighting the core enrichment part to help readers better understand what "core enrichment" means.

We have added in the legend of Fig. 3a: "Genes located prior to the red line represent core enrichment genes".

Reviewer #2 (Comments to the Authors (Required)):

Wei et al define a single-sample differential co-expression method combining the previously described z-score based method for differential co-expression and the rho statistics (Dai et al, 2019) for quantifying co-expression in single cells. Moreover, they add a pathway-enrichment step to select genes with high levels of network rewiring within a highly-rewired pathway.

To assess the reliability of their method, they apply it to two breast cancer datasets, comparing ER+ and ER- tumors and testing the relationship between their differential co-expression score and patients' outcome upon endocrine therapy.

The manuscript is clear and well written, and the topic is highly relevant to the field.

We thank the reviewer for the valuable feedback and recognition of our work.

Major comments:

- Alternative measures of co-expression modules in single samples exist (e.g. module eigengene as defined in WGCNA). How well do they perform in predicting patients' outcome upon endocrine therapy, when based on the selected estrogen response gene set? The authors should show the results using the module eigengene and comment on the differences with their metric.

We appreciate the reviewer's valuable suggestion and have conducted the suggested analysis. As detailed in the newly added paragraph in section 2.5: "In the context of module-based co-expression analysis, the eigengene serves as a commonly used representative expression profile of gene module. It is computed for each gene module by deriving the first principal component from the expression profiles of all genes contained within that module. This approach facilitates the exploration of associations between modules and external variables, such as disease status or treatment response. However, it is important to note that unlike Cosinet, which quantifies variation in co-expression patterns, the eigengene essentially functions as a weighted average expression profile that captures the predominant expression pattern within the module. It provides a summary of the primary variation in gene expression observed within the module across different samples. Consequently, Cosinet provides a measure at the level of gene-gene co-expression relationship that is not present in eigengene. We further evaluated the predictive performance of Cosinet and eigengenes for 5-year clinical outcomes in both datasets. Specifically, we identified a single module for ER+ samples and a consensus module for both ER+ and ER- samples based on the selected estrogen response gene set using WGCNA[Langfelder and Horvath,2008], and then calculated the corresponding module eigengenes. Three-fold cross-validation was performed on each dataset, and the resulting area under the receiver operating characteristic curve (AUC) for the test sets is shown in Fig. S4 (with detailed AUC values provided in Table S2). Our results show that Cosinet outperformed eigengenes in all tasks, with an over- all improvement of 0.09 over module eigengene and 0.1 over consensus module eigengene". And in the Methods section, we added: "We used the survAUC R package (version 1.2.0, <https://cran.r-project.org/package=survAUC>) to compute the AUC for the survival outcome".

- The code for Cosinet should be made available in a public repository, ideally as an R package

We have added in the Data Access section: "The Cosinet R package is publicly available on GitHub (<https://github.com/LanyingWei/cosinet>) with detailed step-by-step instructions and examples".

Minor comments:

- Methods for quantifying co-expression networks in single samples have been developed (e.g. <https://doi.org/10.1093/bib/bbaa268> and reviewed in <https://doi.org/10.1093/bib/bbz089>) and should be mentioned in the introduction.

In the Introduction part, we added: "Recent advancements in this field, exemplified by sample-specific gene interaction perturbations measured by Chen et al.[Chen et al,2021] and co-expression network reconstruction methods on single samples as discussed by Guo et al. [Guo et al, 2020], have underscored the need to examine individual-specific gene interactions for personalized therapy applications".

- In the list of module-based DCE methods, the method developed by Lanciano et al., 2022 is

missing (doi: 10.1093/gigascience/giad010).

We have added the method to the list.

- Figure 3b is not readable.

We have separated Figure 3b as an independent figure to enhance its clarity and readability.

- All R packages employed should be cited

Following the text format guidelines of Life Science Alliance, which specify that references should be limited to articles published or accepted for publication at a named journal or posted on a preprint server, we have carefully cited all R packages for which a published article is available. Additionally, for packages without associated articles, we have included URL links within the text for reference.

- The choice of the two breast cancer datasets (among the hundreds that have been published) should be motivated.

We added in the Data Access section: "We chose these two datasets due to their large sample sizes, recent data (published in 2018 and 2022, respectively), complete and detailed clinical annotations, and consistent data collection, which together provide a strong foundation for comprehensive and up-to-date research on ER+ breast cancer outcomes with hormone therapy."

November 3, 2023

RE: Life Science Alliance Manuscript #LSA-2023-02253-TR

Dr. Yingsheng Zhang
Beijing StoneWise Technology Co Ltd
Danling SOHO, No.6 Danling Street, Haidian District
Beijing
China

Dear Dr. Zhang,

Thank you for submitting your revised manuscript entitled "Patient-Specific Analysis of Co-expression to Measure Biological Network Rewiring in Individuals". We would be happy to publish your paper in Life Science Alliance pending final revisions necessary to meet our formatting guidelines.

- please upload your main manuscript text as an editable doc file
- please add ORCID ID for the secondary corresponding -- they should have received instructions on how to do so
- please remove figures from the main manuscript text
- please upload your Tables in editable .doc or excel format
- please add your main, supplementary figure, and table legends to the main manuscript text after the references section
- please be sure to label the Supplemental Figures file as Supplemental, rather than as Related Manuscript File

A. FINAL FILES:

B. MANUSCRIPT ORGANIZATION AND FORMATTING:

**Submission of a paper that does not conform to Life Science Alliance guidelines will delay the acceptance of your

manuscript.**

The license to publish form must be signed before your manuscript can be sent to production. A link to the electronic license to publish form will be available to the corresponding author only. Please take a moment to check your funder requirements.

Sincerely,

Reviewer #2 (Comments to the Authors (Required)):

The authors carefully addressed all the points I raised.
I therefore consider the manuscript suitable for publishing.

November 6, 2023

RE: Life Science Alliance Manuscript #LSA-2023-02253-TRR

Dr. Yingsheng Zhang
Beijing StoneWise Technology Co Ltd
Danling SOHO, No.6 Danling Street, Haidian District
Beijing
China

Dear Dr. Zhang,

Thank you for submitting your Methods entitled "Patient-Specific Analysis of Co-expression to Measure Biological Network Rewiring in Individuals". It is a pleasure to let you know that your manuscript is now accepted for publication in Life Science Alliance. Congratulations on this interesting work.

DISTRIBUTION OF MATERIALS:

Again, congratulations on a very nice paper. I hope you found the review process to be constructive and are pleased with how the manuscript was handled editorially. We look forward to future exciting submissions from your lab.

Sincerely,
